# Nanopore Technology for the Application of Protein Detection

**DOI:** 10.3390/nano11081942

**Published:** 2021-07-28

**Authors:** Xiaoqing Zeng, Yang Xiang, Qianshan Liu, Liang Wang, Qianyun Ma, Wenhao Ma, Delin Zeng, Yajie Yin, Deqiang Wang

**Affiliations:** 1Chongqing University, 174 Shazheng Street, Shapingba District, Chongqing 400044, China; zeng_xq@foxmail.com (X.Z.); xy@cigit.ac.cn (Y.X.); mawh@cqu.edu.cn (W.M.); 2Chongqing Institute of Green and Intelligent Technology, Chinese Academy of Sciences, Chongqing 400714, China; liuqianshan@cigit.ac.cn (Q.L.); wangliang83@cigit.ac.cn (L.W.); Ang63521elina@gmail.com (Q.M.); zeng_dl@foxmail.com (D.Z.); 3Chongqing School, University of Chinese Academy of Sciences, Chongqing 400714, China; 4School of Pharmacy and Bioengineering, Chongqing University of Technology, Chongqing 400054, China; 5School of Optoelectronic Engineering, Chongqing University of Posts and Telecommunications, Chongqing 400065, China

**Keywords:** nanopore, single-molecule detection, protein detection, solid-state nanopore

## Abstract

Protein is an important component of all the cells and tissues of the human body and is the material basis of life. Its content, sequence, and spatial structure have a great impact on proteomics and human biology. It can reflect the important information of normal or pathophysiological processes and promote the development of new diagnoses and treatment methods. However, the current techniques of proteomics for protein analysis are limited by chemical modifications, large sample sizes, or cumbersome operations. Solving this problem requires overcoming huge challenges. Nanopore single molecule detection technology overcomes this shortcoming. As a new sensing technology, it has the advantages of no labeling, high sensitivity, fast detection speed, real-time monitoring, and simple operation. It is widely used in gene sequencing, detection of peptides and proteins, markers and microorganisms, and other biomolecules and metal ions. Therefore, based on the advantages of novel nanopore single-molecule detection technology, its application to protein sequence detection and structure recognition has also been proposed and developed. In this paper, the application of nanopore single-molecule detection technology in protein detection in recent years is reviewed, and its development prospect is investigated.

## 1. Introduction

The nanopore single-molecule detection technology was originated from the invention of Kurt counter and the recording technology of single-channel current. The single-molecule analysis method was developed in 1996, which is a new sensing detection technology with the advantages of high sensitivity and versatility [1]. Nanopores are mainly divided into two kinds: biological nanopores [2,3,4,5,6] and solid-state nanopores [7,8,9,10]. Biological nanopores are also called transmembrane protein channels [11,12]. The transmembrane protein channels include α-hemolysin (α-HL) [13], Mycobacterium smegmatis porin A (MspA) [14,15], aerolysin (AeL) [16], bacteriophage phi29 motor (phi29) [17], cytolysin A (ClyA) [18] etc., which contain channels of different spatial dimensions. The advantages of the biological nanopores are their well-defined and highly reproducible nanopore size and structure, and the modifiable amino acid residues. However, insertion errors of the biological nanopores are usually caused. The lipid bilayer, the support of biological pores, is not stable enough and is difficult and time-consuming to set up as well. Thus, various solid-state nanopores have been fabricated to solve these deficiencies of the biological nanopores. Various membrane materials are used for solid-state nanopores fabrication [19], such as silicon nitride (Si_3_N_4_) [20], silicon dioxide (SiO_2_) [21], aluminum oxide (Al_2_O_3_) [22], graphene [23], and boron nitride (BN) [24] etc. Solid-state nanopores have many advantages, such as chemical, thermal, and mechanical stability, size adjustability, and integration. However, the solid-state nanopore sizes and structures are not reproducible enough due to the limitation of fabrication techniques. The solid-state nanopores have a lower signal-noise ratio (higher noise) than that of biological nanopores. The principle of the nanopore-based technology is that an electric field is applied to both sides of the nanopores in the buffer solution. An ionic current signal is generated when electrolyte ions move through the pore. The ionic current blockage is observed and measured when the pore is blocked by an analyte. The physical and chemical properties of the molecule such as molecular structures, size, chirality, and net charges, can be calculated by analyzing the ionic current blockades. So far, this technique has been widely used in the fields of gene sequencing [25,26,27,28,29], polypeptide secondary structure [30,31] and protein structure [13,32,33,34,35,36,37] or aggregation state detection [38,39,40], drug screening [41,42,43], molecular interaction [44,45,46], virus recognition [47], and metal ion detection [48,49,50,51].

Proteins are the expression of the genetic code and play an important role in living organisms, controlling the structure and shape of organisms; participating in redox reaction, neuronal transmission, electron transport in cells, enzyme catalysis; and involved in intercellular transport and the recognition and binding of gene expression antigens. The common methods were used for protein secondary structure evaluation, such as X-ray crystallography [52], nuclear magnetic resonance (NMR) [53], Ultrasensitive fluorescence microscope [54], circular dichroism (CD) [55], infrared spectroscopy (IR) [56] and atomic force microscope (AFM) [57], etc. They cannot obtain the dynamic information of protein in real-time. Super-sensitive fluorescence microscopy and atomic force microscopy can detect protein interaction and structure information. However, these methods require chemical modification of the detected molecules, or require large and complex instruments. These instruments tend to be expensive and cumbersome, which may damage the properties of the proteins themselves, and complex experimental procedures may bring more uncertainty. Compared with traditional methods, the nanopore technology has significant potential in the detection sensitivity timeliness and equipment operation convenience, etc. In this paper, the nanopore single-molecule detection technology application in protein detection was summarized. We mainly discuss the latest progress of nanopore technology in protein detection, as the following aspects: (1) Detection of conformational changes in protein space; (2) Protein–protein interaction detection; and (3) Protein–DNA interaction detection. It is expected to provide technical reference for promoting the development of gene therapy technology and precision medicine plan in the future and triggering a new revolution in medicine.

## 2. Detection of Conformational Changes in Protein Space

The spatial conformation or size of a protein has a close influence on its function. The characteristic shift signal is generated when the substance passes through the nanopore, which can reflect the relevant information such as the size and shape of the charge and amino acid sequence of the molecule. The folding of linear polypeptides into three-dimensional structures is a critical step in the formation of proteins. Misfolding may lead to irreversible biological consequences, such as disease. Up to now, many researchers have been studying diseases caused by protein misfolding. For example, the formation of amyloid fibrils associated with Alzheimer’s disease (AD) by peptide misfolding and aggregation [58,59]; Parkinson’s disease (PD) associated with Lewy bodies composed of α-synuclein [60,61]; and Extensive Transmissible Spongiform Encephalopathy (TSEs) caused by foreign or their own genetic mutations, causing human normal type C prion protein (PrPc) to misfold into type SC prion protein (PrPsc) [62,63]. However, the dynamics of protein folding and unfolding are complex and difficult to study. The emergence of nanopore technology has laid an important research foundation for solving this problem. Nanopore technology is characterized by ultra-sensitive, label-free, and high throughput. The folding/unfolding process of a single protein with different structures can be dynamically observed simultaneously. The folding and unfolding mechanisms of individual proteins can be better understood by the duration of the feature and blockade current.

### 2.1. Characterization of Protein Folding/Unfolding

In living organisms, the folding of a specific sequence of amino acids into a three-dimensional structure is a key process that produces specific functions of proteins and determines their conformations [64]. A protein is a folded polymer with a complex free energy structure, including many transition and conformational substrates [65]. Folding is a complex distribution process that directs proteins to their lowest and most stable natural state [66,67,68,69]. Through computer simulation, Freedman et al. found that in the nanopore translocation experiment, highly charged BSA molecules may undergo partial denaturation under the action of voltage when entering the nanopore [70]. Then, a nanopore detection method for protein folding/unfolding structure was proposed [71]. That is to say, under an electric field greater than 10^6^ V/m, the protein folding/unfolding detection method was controlled by changing the applied voltage, as shown in Figure 1. Different characteristic blocking current and resident time will be generated when the three substances to be measured pass through the nanopore at different voltages. These data and the change of Gibbs free energy could be used to show that the unfolding of the protein was not direct as the denaturant urea, but gradually unfolding with the increase of voltage. By comparing the conformational changes of two kinds of mutated and unstable proteins, it proved that voltage affects the stability of protein structure. It was further demonstrated that nanopores can distinguish the folded and unfolded conformations of proteins. Fologea et al. used protein bovine serum albumin (BSA) as a model to discuss the current block and the amplitude and duration associated with the protein unfolding process, using the method to identify the fold of bovine serum albumin (BSA) and the folding conformation [72]. They used denaturant guanidine hydrochloride, urea, and sodium dodecyl sulfate (SDS) and dithiothreitol (DTT) in different temperatures on fat protein denaturation. The occupied or excluded volume of the denatured protein molecule in the nanopore depended on the conformation or shape of the protein.

The folded protein had a larger repulsive volume in the nanopore, so its unfurled form impedes more ionic current flow and produces larger current blocking amplitude. Protein in the nanopore also depended on the residence time in the state of protein folding through the relative current decline amplitude Δ*I_b_*/*I*_0_, to characterize unfolding protein. The conformations of the BSA molecule in the solid nanopore were measured in folded and fully unfolded parts, as shown in Figure 2. Also, Oukhaled et al. studied the folding and stability of different proteins by combining nanopore and electrical detection [73]. The results showed that the fully developed protein only caused a short blocking current, and its frequency increased with the increase of denaturant concentration or temperature, showing an S-shaped denaturation curve. 

The partially folded protein was blocked in the nanopore for a long time, but the duration was gradually decreased with the increase of denaturing agent concentration. Rodriguez-Larrea and Bayley used biological nanopores to simulate how folded proteins moved inside and outside the cell using narrow transmembrane (phospholipid bilayer) pores [74], as shown in Figure 3. Since proteins must be unrolled during the process of movement, the team labeled the protein substrate with oligonucleotides and drove thioredoxin through the nanopore in a four-step translocation mechanism. First, DNA tags were trapped by the nanopore. Second, the oligonucleotides were pulled through the pore, causing the C terminus of thioredoxin near the pore entrance to unfold locally. Third, the rest of the protein unfolded spontaneously. Finally, the developed peptide diffused through the nanopore to the receptor compartment. This study was different from the mechanism that the protein gradually unfolded through the pore produced in the denaturation experiment of solid nanopores in solution. The following year, this team showed that protein co-translocation was related to the sequence of peptide terminus entry in vivo [75]. When thioredoxin was pulled from the N terminal, some molecules unwrapped rapidly, while others from the C terminal were 100 times slower. This rapid and highly sensitive characterization of protein structure was expected to provide an effective reference for us to understand the unfolding mechanism of biological proteins and the application and modification of protein detection by nanopore technology. 

### 2.2. Characterization of Protein Size and Shape Aggregation

The three-dimensional structure of proteins plays an important role in the protein dynamics of human biological systems. Until now, characterizing and quantifying the shape of proteins at the single molecular level has been a challenge. Nanopore, as a novel single-molecule sensor, is expected to be used as a novel protein fingerprint recognition method for single protein fingerprint recognition. Fologea et al. compared BSA with larger protein fibrinogen to investigate the effect of protein size and structure on blocking signals [76]. With average current blocking amplitude Δ*I_b_*, shift time *t_d_*, and integral area of block *A_ecd_*, they estimated the relative charge and size of protein molecules. This nanopore technology could measure the properties of individual protein molecules sequentially and determined the distribution of these properties in real-time under natural conditions. It was also shown that solid nanopores could be used to characterize unknown proteins with known charge conformations and the size of labeled proteins. Sha et al. characterized and distinguished spherical and non-spherical proteins by kinetic binding experiments of detecting proteins in solid-state nanopores [77]. In addition, they differentiated the long shape of BSA and spherical Con. A by relative block current *(*Δ*I_b_*/*I*_0_*)*. At low voltages, the two proteins had only one blocking current, while at voltages greater than 300 mV, the non-spherical protein BSA had two obvious blocking currents. It was found that when the voltage was higher than a certain threshold, the non-spherical protein BSA showed two orientations through the pore instead of unrolling the protein molecule, thus generating a two-stage blocking current. For spherical protein Con. A, the average cross-sectional area of the protein was not changed under high and low voltages so that its cross pore orientation was not affected, and the ionic current was not affected. This point was also verified by the experiment step. Giamblanco et al. explored the potential of nanopores in detecting protein aggregation [78]. Three model proteins (mutated lactoglobulin, lysozyme, and bovine serum albumin) were selected because of their different morphologies (protofilaments or spheres) during the early stages of aggregation. Also, the nanopores were modified with polyethylene glycol (PEG). The results showed that the modified nanopores could effectively prevent the non-specific adsorption of protein aggregation, prolong their service life, and clearly distinguished the aggregation morphology of protein, as shown in Figure 4. Wei et al. observed reversible binding and de-binding of proteins to receptors in real-time, and interaction parameters were statistically analyzed from single molecular binding events [79]. 

To demonstrate the generality of this approach, the His-labeled proteins were detected and the rodent immunoglobulin G (IgG) antibody subclasses were distinguished. Yusko et al. used nanopore to determine the approximate shape volumic-charge rotational diffusion coefficient and the dipole moment of a single protein in real-time [80]. Thus, a theory was developed to quantitatively understand the regulation of ionic current resulting from the rotational dynamics of a single protein as it passes through an electric field in a nanopore. The results indicated that they could be used to identify, characterize, and quantify proteins and protein complexes, and those had potential implications for biomarker detection of structural biology proteomics and routine protein analysis. 

### 2.3. Characterization of Induced Protein Conformational Changes

The charge and structure of proteins were easily affected by some ions or ligands at pH value, so they can play corresponding biological functions by changing their conformation. Saharia et al. studied the response of human serum transferrin (hSTf) to pH and voltage by using silicon nitride nanopores [81], shown in the Figure 5. At pH > PI, hSTf mainly existed in the form of pure folded (*holo*) rich in Fe(III). At pH < PI, it mainly existed in the form of leaflet opening (*apo*) without Fe(III). The translocation of hSTf was similar to that of electrophoresis; when the voltage increased, the kinesin gradually unfolded. Because when proteins broke down under high pressure, pores inside them were exposed, resulting in smaller molecules. The results showed that the nanopore could distinguish the hSTf in the form of *holo* and *apo* in the mixed solution (pH 8) and could analyze the folding and unfolding of proteins in a certain pH value and applied voltage range. HSTf was responsible for transporting insoluble iron to cells, which was very important for iron homeostasis in the human body. These nanopore-based methods have the potential to detect the abnormal presence of specific proteins and to study the structural and kinetic properties of proteins, which have great application prospects in clinical detection. 

Waduge et al. found a close correlation between the blocking current distribution shape and protein fluctuations through molecular dynamics simulations [32]. They also studied the conformational change of calmodulin from a calcium-free structure to a calcium-carrying structure by using a nanopore composed of silicon nitride and HfO_2_. After characterizing calmodulin without calcium, they added CaCl_2_ to the sample pool and observed significant changes in residence time and blocking current, which might be related to changes in calmodulin configuration. Hu et al. also used molecular dynamics (MD) simulation to study the translocation of calmodulin silicon nitride nanopores [82]. Calmodulin was first fixed in the nanopore, and the two types of calmodulin had obvious blocking ion current. Then, in the translocation simulation, appropriate voltage and similar pore sizes were selected for the proteins, and it was easy to find that there was a significant difference in temporal resolution between the two states of calmodulin. 

Chae et al. effectively detected the complex conformational changes of protein–protein interaction (PPI) by using solid nanopore [83], which was a novel and versatile drug screening method for various PPIs. To effectively detect the conformational change induced by PPIs, they designed the fusion protein MLP (MDM2-p53TAD), in which MDM2-p53TAD was linked by 16 amino acids. The spherical conformation of MLP was characterized by unimodal translocation events, while the dumbbell conformation of Nutlin-3 binding MLP was characterized by bimodal signals. With the increase of Nutlin-3 concentration, the ratio of bimodal and unimodal signals increased from 9.3% to 23.0%. The migration dynamics of the two different MLP conformations under different applied voltages were analyzed. Further analysis of the fractional current in the bimodal signal peak to explore the structure of the protein complex was designed. This nanopore sensing method could be widely used in the screening of PPIs inhibitors and the study of protein conformation. In the following 2 years, they further examined drug-induced conformational changes in the p53-Mdm2 protein complex using nanopore [84]. A p53 peptide was designed to link MDM2 complex protein, which was linked by six amino acids. By inhibiting the interaction between the p53 peptide and MDM2, it changed from a spherical structure to a dumbbell structure. In NMR experiments, no significant crossover was observed after the addition of Nutlin-3. However, with the addition of Nutlin-3, the nanopore experiment clearly showed a bimodal signal. The bimodal fraction observed in the unimodal signal increased from 8.77% to 22.03%, while the concentration of Nutlin-3 increased from 1 to 10 times the molar ratio. From the nanopore data, the residence time of Nutlin-3 binding protein in the extended form was estimated, which was longer than that in the spherical form (two times). Finally, the hydrodynamic diameters of the local peaks of the bimodal signals were calculated and compared with the results of X-ray crystallography. This method demonstrated the feasibility of nanopore detection to verify protein conformational changes at the single molecular level by inhibiting protein–protein interactions. It is hoped that this method can be applied to the development of a drug screening platform for detecting conformational changes induced by PPIs. In conclusion, some progress has been made in protein characterization and protein conformation detection by solid-state nanopores, which provides a basis for disease research and drug screening. 

## 3. Detection of Protein-Protein Interactions

Gene determines the protein, and the interaction between proteins determines the main function, so the interactions between proteins are the basis of cell life activities. The study of protein interactions looks at their corresponding biological functions, such as molecular linkage, affinity, binding kinetics, and binding-induced folding kinetics. Also, the identification of protein-conjugated conjugates plays an important role in the field of biological sciences, such as qualitative and quantitative analytical tests and diagnostics for a variety of different biological species. Therefore, it is of great significance to study the antigen–antibody and ligand–receptor interaction by using nanopore detection technology.

In 2006, Uram et al. used submicron pores for the first time to detect characterization and quantification of the binding of polyclonal antibodies to intact paramecium chlorella virus (PBCV-1) particles [85]. This method could detect the formation of viral aggregates quickly and without labeling and without fixing or modifying antibodies or viruses. The maximum number of antibodies that could bind to the individual virus particles is approximately 4200. This method presented some challenges in detecting small numbers of antibodies or other molecules. Based on this study, in 2013, Freedman et al. used solid-state nanopores to detect HIV envelope glycoprotein gp120, antibody gp120, and its antibody coupling complex and related other complexes [86]. The results showed that the probability density of the monomer and dimer of gp120 antibody had a bimodal signal, and there were univalent and polyvalent binding states between gp120 and the dimer of gp120 antibody, and the univalent binding states were dominant. Also, the team added bovine serum protein (BSA) and fetal bovine serum (FBS) interferers to the gp120 antigen-antibody complex to verify that the interferers did not bind to the antibodies, and the binding signal could be isolated from the background. This study demonstrates that unlabeled nanopore detection can be used to qualitatively detect antibodies and their antigen-conjugated complexes, providing a new method for other antibody detection and drug screening design. 

In 2016, Kwak et al. used solid-state nanopore to study the interaction between the anti-cancer therapeutic p53 transactivation domain (p53TAD), MDM2, and its inhibitory effect on Nutlin-3, a small-molecule MDM2 antagonist [87]. When p53TAD and MDM2 were detected separately, it was found that the two proteins had opposite charges in the neutral buffer and formed a complex with negative charges, which could prevent MDM2 translocation. However, translocation signals appeared when Nutlin-3 was added, because Nutlin-3 destroyed the MDM2-GST-p53TAD complex and re-released MDM2. In summary, the results indicate that solid-state nanopore detection technology can establish a platform for target protein–protein interaction drug target screening based on the advantages of the unlabeled ultra-sensitive and low detection limit. High-throughput detection of transient protein–protein interactions is a challenging experiment. 

In 2019, Avinash and Kumar used biological nanopores to monitor protein–protein interactions in real-time [88], as shown in Figure 6. In complex molecules containing FBS, the binding and release of protein ligands to receptors could be clearly distinguished using this nanopore technique. In the same year, Chuah et al. used silicon nitride nanopores to detect prostate-specific antibody (anti-PSA)-modified nanoparticles and PSA captured by this magnetic bead modifier [3]. In a free magnetic field, the nanoparticle trapped the analyte and PSA with diffusing. If the PSA molecule was trapped by a magnetic bead, it could not be removed when the magnetic field was reversed, but unbound PSA magnetic nanoparticles could be removed, which could avoid false signals. When the array nanopores were blocked, it could be seen that the blocking current was longer. The concentration of PSA was measured with whole blood samples and the detection limit was the flying-Moore level. This study breaks the conventional detection techniques, replaces diffusion with active capture, eliminates false signals, improves specificity and detection limit, and provides important basic research for quantitative detection of various proteins or nucleic acids. 

## 4. Detection of Protein-DNA Interactions

The interaction of proteins with specific DNA sequences is critical in the control of gene expression and the regulation of replication. The monomolecular approach provides an excellent ability to uncover the mechanisms and dynamics of these interactions. As biological macromolecules, protein and DNA play an important role in the structure and function changes of living organisms due to their mutual reactions. 

In 2009, Dekker’s team reported the force spectrum analysis of RecA protein wrapped DNA molecules by solid nanopores with optical tweezers, as well as the translocation analysis of solid nanopores [89,90], as shown in Figure 7. The experimental results showed that the translocation blockade time was three orders of magnitude, and the blocking value had two different regions. It was believed that the large blocking value was caused by the RecA-dsDNA translocation, and this result was confirmed by the optical tweezers system. Also, different applied voltages were applied to distinguish different translocations: At voltages below 150 mV, the rate of translocation events increased with increasing voltages; however, the translocation rate remained the same when the voltage was higher than 150 mV. The research has potential in basic science and genome screening. In 2010, the team further examined both RecA-DNA and bare DNA using solid-state nanopore translocations that produce significantly different current blocking signals, in which the RecA-DNA strand consists of 5 RecA proteins bound to 15 base pairs of DNA [91]. These results demonstrate that it is possible to use solid-state nanopores to read information along with DNA at high resolution, a step towards genomic screening. 

In 2011, Spiering et al. studied the dynamics of translocation of single protein molecules attached to double-stranded DNA using nanopore force spectroscopy [92]. By modeling and measurement, it was found that the different asymmetry and delay force signals depended on the elasticity of protein charge DNA and its counterion screening. The model results were in good agreement with the measured force curves. The force had a linear relationship with the applied voltage and showed a small blocking phenomenon in the violent random movement back and forth. Ledden et al. studied the translocation time of DNA and protein molecules in solid nanopores [93]. The results showed that the translocation of biopolymers through nanopores depended on the properties of the polymer, including its conformational state size, conformational charge, and charge distribution. The natural state of protein and DNA translocations approximately followed simple one-dimensional biased diffusion of charged particles. Due to the polypeptide’s heterogeneous charge sequence, the undeveloped proteins were subjected to sequence-specific coupled electrophoresis and thermal activation processes. In 2012, Raillon et al. used solid-state nanopore sensing technology to detect a single *Escherichia coli* RNAP-DNA transcription complex and a single *Escherichia coli* RNAP enzyme to study the interaction between a single protein and nucleic acid [94]. These two types of molecular translocations and naked DNA translocations were distinguished and identified according to their specific conductance translocation characteristics. This provides a new perspective for the study of the transcriptional process at the single molecular level. In 2014, Meervelt and Soskine used biological nanopores to detect two isomer binding configurations in protein–aptamer complexes [95]. In 2016, Kong et al. designed DNA vectors as a new method for specific protein detection based on nanopores [96]. In the system (which used the standard biotin-streptavidin and digoxin anti-digoxin binding systems), the target protein molecule bonded to a defined location on the DNA strand, causing a secondary transient current drop when the DNA was translocated. DNA was incubated with different concentrations of target proteins before detection, and the results showed that protein concentrations could be quantified within the nanoscale concentration range. This study demonstrates the potential of a new quantitative and specific protein assay protocol using DNA carrier methods. In 2017, Sze et al. used aptamer-modified DNA carriers to screen monomolecular composite nanoporous proteins in human serum [97]. 

Celaya and Perales-Calvo developed a nanopore method to distinguish naked DNA from DNA–protein complexes [98], which tests the inhibitory effect of small molecules on complex formation and the mechanism of action and facilitates the development of transcription factor binding inhibitors. In 2019, Kaur et al. detected the binding RNA polymerase (RNAP) on 48.5 kbp (16.5 μm) λDNA using silicon nitride-based nanopore [99]. To prevent RNAP from separating from λDNA molecules in the nanopore at high electric fields, the team used formaldehyde to cross-link RNAP proteins to λ-DNA. The binding efficiency and binding position of RNAP on λ DNA were estimated by analyzing the duration and amplitude of current blocking signals of translocation events and secondary translocation events. This is shown in Figure 8. The results showed that a single RNAP binding site was indistinguishable from the nanopore itself; the binding efficiency of RNAP/λDNA was about 42% under the experimental condition of 6 to 1; the binding sites of RNAP were mainly at 3.51 ± 0.53 μm, most likely corresponding to the strong promoter regions at 3.48 μm and 4.43 μm (38 kbp and 35.6 kbp) on λDNA. The work provides a new perspective and complexity for studying the binding of the transcription factor RNAP at different locations on very long DNA molecules. In 2020, Chau et al. enhanced the detection of DNA and protein by using macromolecule aggregation in solid-state nanopores [13]. In conclusion, nanopore detection technology provides an important basis for the development of gene sequencing drug screening inhibitors and the application of medical diagnosis. 

## 5. Conclusions and Outlook

Nanopore detection technology is a new type of nanoscale electrical sensor that has many advantages such as fast, unlabeled, and high sensitivity. In this paper, the applications of this technique are reviewed in the fields of conformational changes in protein space, protein–protein interactions, and protein–DNA interactions, and provide major opportunities for biological and disease research. It plays an important role in promoting the development of gene therapy technology in medical diagnosis and future precision medicine plans and triggering a new revolution in medicine. 

Although the application of this technology is promising, difficulties still need to be overcome to achieve better progress and breakthroughs. How to improve the temporal and spatial resolution of the analyte through the nanopore so that it can be more controlled and more realistic to reflect the properties of the protein, is an important topic of interest. To improve the temporal resolution, it can slow down the analyte translocation speed or increase the collected data bandwidth, but at the same time, it will introduce more noise signals. To reduce the noise signals, it is necessary to improve the amplifier selection requirements or improve the nanopore. To improve the spatial resolution, it is necessary to improve the reproducibility of hole making and select the appropriate aperture size and thickness. However, when the thickness becomes very thin, there is a risk of rupture. Obtaining instantaneous spatial conformational changes of individual proteins remains a huge challenge. Although it has been reported that 2D or 3D self-assembled DNA structures have great potential to improve the performance of nanopore detection, a lot of research work is still needed to better combine DNA-assembled nanostructures with nanopores. Alternatively, fluorescence imaging technology can be combined with nanopore current signals to realize photoelectric joint detection, and the instantaneous state of proteins in nanopores can be revealed by the kinetic or physical parameters obtained. Also, if the nanopore detection technology is to be commercialized or productized, it needs to be industrialized, and the size, shape, structure, and surface characteristics of the nanopore should be strictly unified. In conclusion, nanopore detection technology is a promising biosensor with great room for improvement in the field of biology and medicine and has immeasurable application value. Also, if combined with other detection techniques, it is more conducive to the identification of proteins and the realization of high-throughput protein sequencing, which will bring revolutionary changes to proteomics research. 

## Figures and Tables

**Figure 1 nanomaterials-11-01942-f001:**
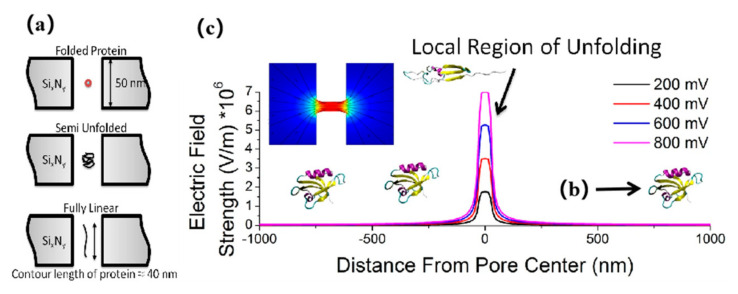
(**a**) Schematic showing the expected protein states with the principal sensing element, a 15-nm solid-state nanopore, drilled in a 50-nm-thick silicon nitride membrane. (**b**) Crystal structure for the SAP97 PDZ2 domain. The Protein Data Bank (PDB) CODE: 2X7Z. (**c**) Finite element simulations showing the electric field plotted as a function of distance from pore center; pore dimensions: 15 nm diameter, 50 nm thick membrane. This was performed over the range of 200–800 mV in which the effects on protein folding are investigated [71]. Copyright © 2013, Macmillan Publishers Limited. All rights reserved.

**Figure 2 nanomaterials-11-01942-f002:**
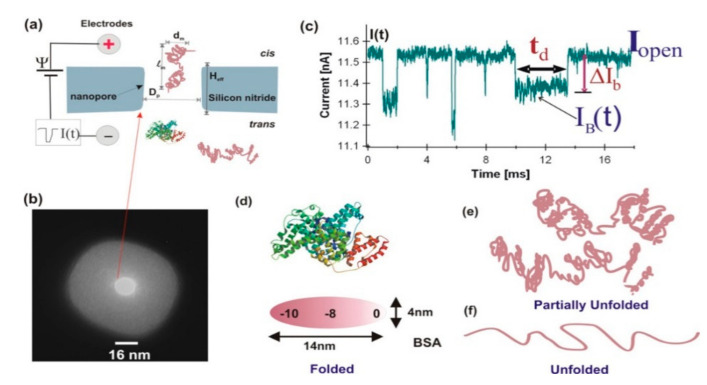
(**a**) Schematic diagram of a nanopore experiment setup. (**b**) A TEM image of a ~16 nm pore used for BSA measurement. (**c**) Several recorded BSA current blockage events in partially denatured conditions (SDS + DTT + 45 °C at pH 7 and 1 M KCl) measured with the nanopore shown in b. (**d**) Illustration of one of the possible conformations of a BSA protein at native state (PDB, 3v03). (**e**) Possible partially denatured form of BSA. (**f**) Completely unfolded form of a BSA molecule [72]. Copyright © 2013, Macmillan Publishers Limited. All rights reserved.

**Figure 3 nanomaterials-11-01942-f003:**
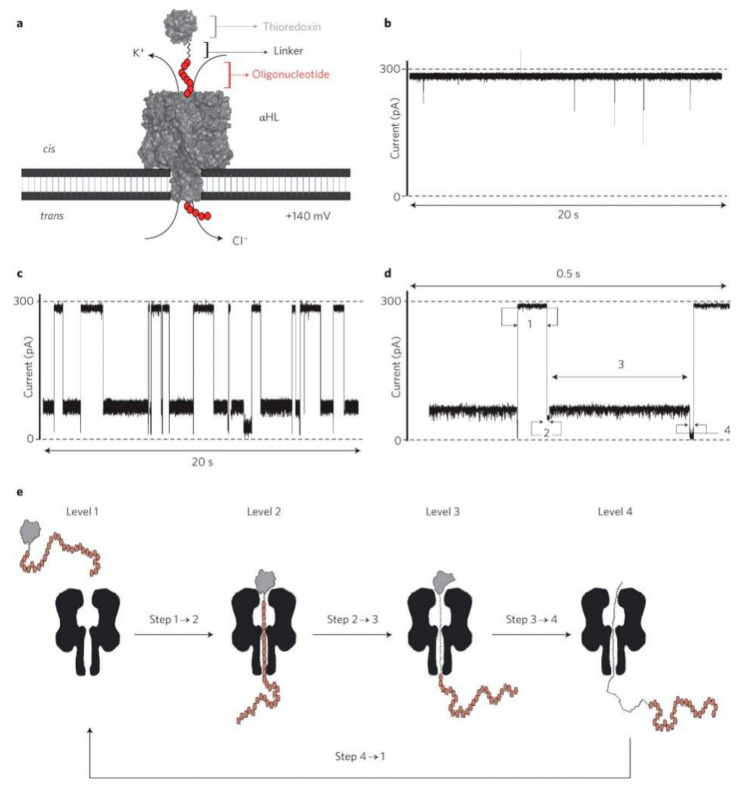
Interaction of V5-C109-oligo (dC)_30_ with the αHL pore. (**a**) The pore is inserted in a lipid bilayer from the *cis* compartment and a potential is applied to cause an ionic current to flow through the pore. (**b**) Current trace at +140 mV in 2 M KCl in the absence of V5-C109-oligo (dC)_30_. (**c**) Current trace at +140 mV in 2 M KCl with V5-C109-oligo (dC)_30_ (0.4 µM, cis). (**d**,**e**) Level 1: V5-C109-oligo (dC)_30_ is in solution and the pore is unoccupied. Level 2: The oligonucleotide threads into the pore and pulls on the protein. Level 3: the pulling force causes partial unfolding, allowing the oligonucleotide to traverse the pore and the unfolded segment of the polypeptide to enter. Level 4: The remainder of the polypeptide unfolds spontaneously, diffuses through the pore, and leaves through the trans entrance [74]. Copyright © 2013, Macmillan Publishers Limited. All rights reserved.

**Figure 4 nanomaterials-11-01942-f004:**
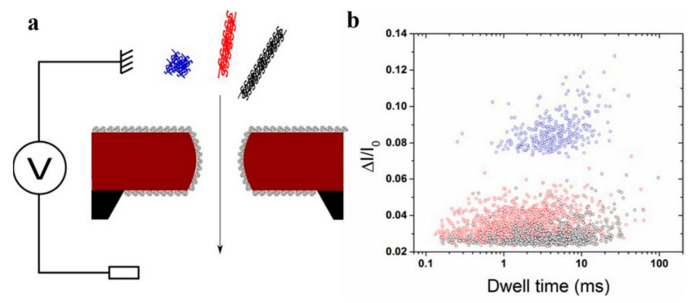
(**a**) Illustration of protein aggregate detection using a nanopore. (**b**) Map of events recorded for β lactoglobulin (red), lysozyme (gray), and BSA (blue) aggregate [78]. Copyright © 2018 Elsevier B.V. All rights reserved.

**Figure 5 nanomaterials-11-01942-f005:**
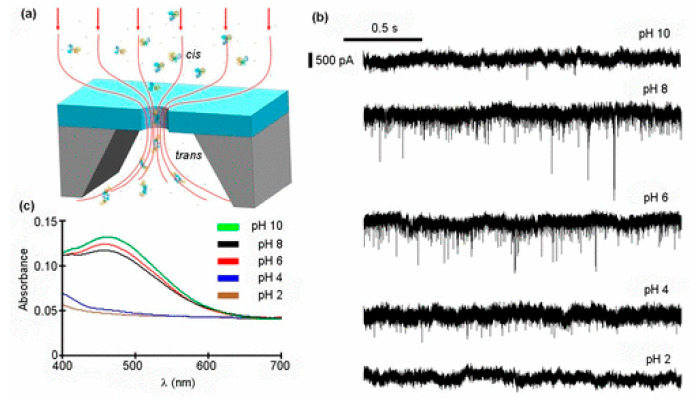
(**a**) Typical setup used for nanopore experiments in buffered (10 mM HEPES) 2 M KCl with low pass filtering at 10 kHz and data acquisition at 200 kHz. The hSTf (250 nM) protein molecules are driven across a ~20 nm diameter nanopore by an applied transmembrane potential. The voltage bias (100−800 mV) is applied to the trans chamber, and the analyte is added to the cis chamber. (**b**) Representative 2 s current traces, from top to bottom, for pH 10, pH 8, pH 6, pH 4, and pH 2 at +800 mV applied voltage. All traces were obtained using the same ~20 nm diameter pore. (**c**) UV−vis spectra of a ~25 μM hSTf solution at pH 10 (green), pH 8 (black), pH 6 (red), pH 4 (blue), and pH 2 (brown) [81]. Copyright © 2019, American Chemical Society.

**Figure 6 nanomaterials-11-01942-f006:**
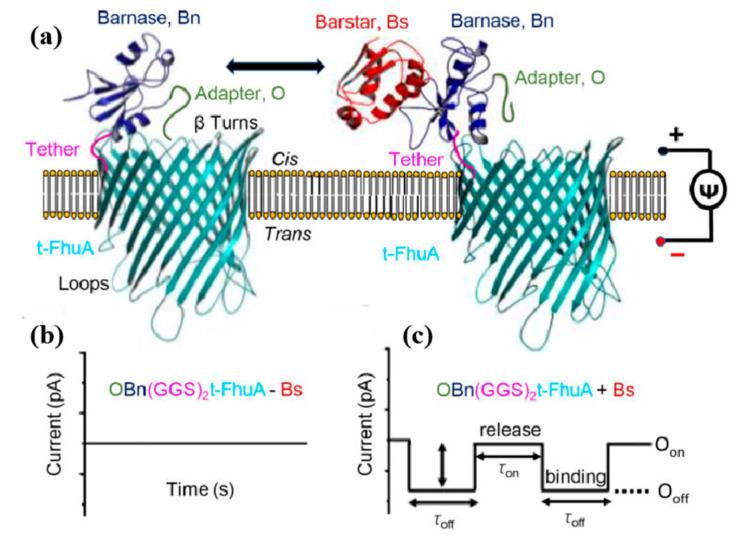
Measuring high-affinity PPIs using a nanopore sensor. A protein pore-based nanostructure for the real-time sampling of transient PPIs is shown in (**a**). The single-polypeptide chain protein comprises a t-FhuA protein pore scaffold, a flexible (GGS)_2_ tether, a barnase (Bn) protein receptor, and an O peptide adapter. The schematic model was created in PyMol using the PDB files ID: 1BY3 (FhuA) and 1BRS (Bn-Bs). Stochastic sensing of transient PPIs using a single OBn(GGS)2t-FhuA protein is shown in (**b**,**c**). The protein nanostructure maintains a basal open-state conductance (**b**). When added to the *cis* side, the Bs protein–ligand is expected to produce current transitions between two conductance substates [3] Copyright © 2018 Nat Biotechnol.

**Figure 7 nanomaterials-11-01942-f007:**
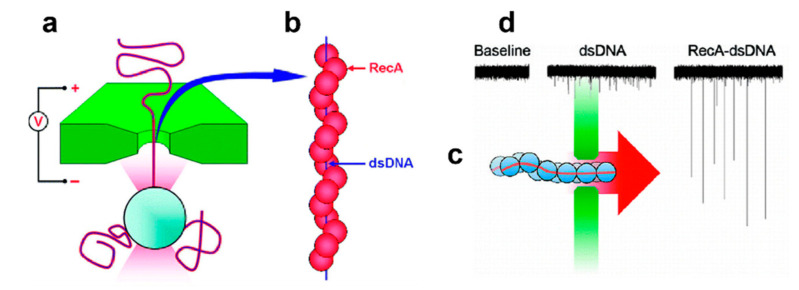
(**a**) Schematic of the experiment, showing a bead-conjugated molecule electrophoretically captured in a nanopore [89]. (**b**) Representation of the RecA-dsDNA filament, where RecA proteins bind along the entire length of dsDNA [89]. (**c**) Schematic of the experiment, showing the nanopore between two liquid compartments. We apply an electric field across the nanopore and measure the resulting current. Molecules are added to a single liquid compartment and translocate through the nanopore toward the positively biased electrode [90]. (**d**) Current recording of a 31.1 nm diameter nanopore at 120 mV before and after the addition of bare λ-DNA or 5 kbp dsDNA/RecA filaments to the negatively biased electrode. Time and current scales as well as the addition of the molecule are indicated. Clear current spikes from the baseline appear upon the addition of the molecules, with much larger current blockades for the RecA-dsDNA filaments as compared to the bare dsDNA [90]. Copyright © 2009, American Chemical Society.

**Figure 8 nanomaterials-11-01942-f008:**
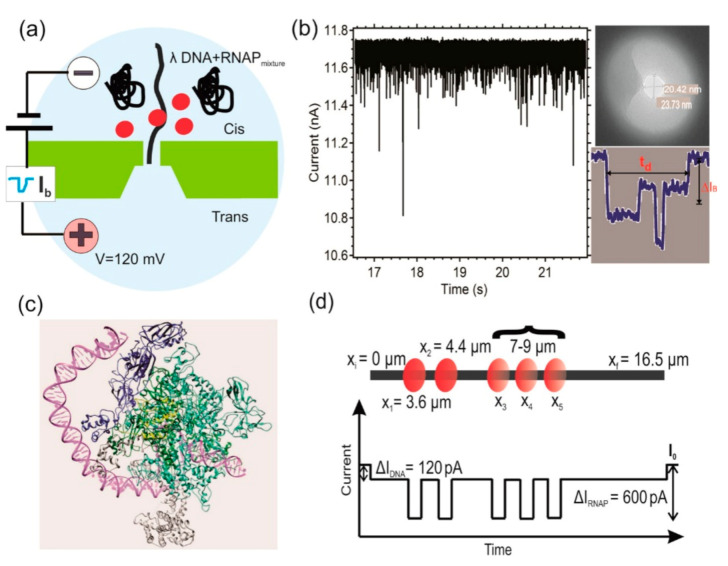
(**a**) Schematics of silicon nitride nanopore immersed in 1 M KCl salt solution with a mixture of λ-DNA and RNAP. (**b**) Trace of current blockage events recorded from a λ DNA and RNAP mixture. The side panel shows a TEM image of 20.42 nm by 23.73 nm size nanopore used in this experiment and an example of current blockage event recorded. (**c**) 3D representation of E. coli RNAP-DNA complex (3IYD pdb file). Two α subunits are shown in blue; subunits β and β′ are in dark green and light green, respectively; ω is in yellow; σ70 factor is shown in white; and purple reflects DNA strand. (**d**) Schematics of RNAP bound on five possible positions on a 16.5 μm (48,502 bp) long λ DNA and below that a predicted current blockage event from a λ-DNA*RNAP complex. The current blockage level due to DNA and RNAP is estimated by ionic volume exclusion caused by DNA and RNAP through a nanopore of given dimensions [99]. Copyright © 2009, American Chemical Society.

## Data Availability

Not applicable.

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
