# Peer review of "Nanopore Technology for the Application of Protein Detection"

_nanomaterials, 2021, doi:10.3390/nano11081942_

Round 1

Reviewer 1 Report

This review paper wants to report on recent results on nanopore technology for protein detection. 

In particular the authors focus the discussion on some major points: (i) the detection of conformational changes; (ii) detection of protein folding/unfolding; (iii) protein size and shape aggregation; (iv) protein conformation changes; (v) protein protein and protein-dna interaction

there are others review papers on nanopore for protein analysis, anyway the structure of this manuscript makes it rather new and from some points of view interesting.

Anyway, in the present form it cannot be accepted for publication.

first of all, the quality of the english is very bad. a large number of grammar errors can be found along the text and them make the reading very hard.

Then the discussion, mainly in the first part is confuse and badly written.

for example - line 46 "Biological nanopores, also called transmembrane protein channels, can embedded in a lipid bilayer membrane or other polymer films, such as α-hemolysin...etc"...reading like this it sounds that a-hemolysin is the lipid membrane!!

in the whole first page of introduction there is no references!! really weird,  considering the huge amount of papers that could be cited on this topic.

line 72: please define REDOX...

line 106-111: "However, by the complexity of protein folding and unfolding
 dynamic process, such studies are still difficult nanopore technology as an
 ultrasensitive unmarked and high flux alternative, can at the same time for different  structure of a single protein folding/unfolding process in dynamic observation  Through the characteristics of the current duration and choke, can better understand a  single protein folding and unfolding mechanism." Impossible to understand!

line 273: "novels"??

line 317: "The maximum number of antibodies that could bind
to the virus approximately 4200"??  .......is 4200?

line 424: "the binding sites of RNAP were mainly at 3.51 μm and 0.53 μm, most likely corresponding to the strong promoter regions of 3.48 μm and 4.43 μm λDNA." not clear at all

the quality of the figures is also very bad. being a review paper it is very important to use high-quality and high-readable figures. moreover, I recommend the authors to include additional figures. 4 figures in total for a 24 pages review is a very low number.

Again, taking into consideration that this paper wants to be a review on the topic, the authors must improve significantly it. moreover, it is fundamental to mention several recent important papers on the topic that are now not mentioned giving the impression that the authors are not up to date.

Some important papers to be cited (mainly in the intro, but also along the different sections)

https://doi.org/10.1038/s41587-019-0345-2

https://doi.org/10.1038/nbt.4316

https://doi.org/10.1038/s41565-019-0549-0

https://doi.org/10.1021/acsnano.9b05156

https://doi.org/10.1016/j.tibs.2019.09.005

 https://doi.org/10.1002/smll.201900036

https://doi.org/10.1002/anie.202000489

https://doi.org/10.1021/acssensors.9b00848

https://doi.org/10.1038/s41587-019-0401-y

https://doi.org/10.3390/ijms21082808

https://doi.org/10.1038/s41578-020-0229-6

https://doi.org/10.1002/smtd.202000356

https://doi.org/10.1088/1478-3975/ab0792

https://doi.org/10.1007/s12274-020-3095-z

https://doi.org/10.1039/C9NR09135A

Reviewer 2 Report

The paper presents a review of the nanopore technology for the application of protein detection. The presentation of methods and scientific results in the current form is not satisfactory for publication in the Nanomaterials journal. The minor and major drawbacks to be addressed can be specified as follows:

  1. Page 1. Are the authors experts on the reviewed subject matter? Which references are their authorship?
  2. Page 2, Abstract. The abstract is very poorly written. Unclear. Too general. It is needs improvement. So are language and grammar errors; for example, the word "detection" appears three times in the last sentence.
  3. Page 3 and others. Why enter shortcuts if they are not used in the reviewed manuscript?
  4. Page 3. Why is there no reference in this part of the introduction?
  5. (i) Page 6, line 132. Fologea et al. ---> Li and Fologea. (ii) Page 7, line 152. Oukhaled et al. ---> Oukhaled and Pastoriza-Gallego. (iii) Rodriguez-Larrea et al. --->Rodriguez-Larrea and Bayley. Check all the manuscripts!!! See the general scheme below:

[i] R. Smith ---> Smith [i]

[ii] R. Smith, A. June ---> Smith and June [ii]

[iii] R. Smith, A. June, D.G. May ---> Smith et al. [iii]

  1. Page 7, line 150. I0?
  2. Page 8, Fig. 3. Inferior drawing quality!!!
  3. Page 11, line 246. (i) Ph ---> pH. (ii) PI?
  4. Page 11, line 254. HSTf ---> hSTf.
  5. Page 12, line 284, "In the following two years, the team further". The team??? What a team?
  6. Page 16, lines 400 and 402. In 2014. [67] Meervelt ---> In 2014 Meervelt.
  7. Page 17, figure captions. Why are copyright consents appear four times?
  8. References are not always in the sentence where they should be.
  9. The topic is quite popular. This work doesn't have the latest news - there are too many older works - earlier than 2015.

Round 2

Reviewer 1 Report

The authors have significantly improved the quality of the manuscript.

now it is almost suitable for publication. I just recommend an additional deep check of the english grammar, considering that some sentences are still hard to be read and understood.

The number of cited paper increased from about 40 to 100, demonstrating that something was missing in the previous version.

Reviewer 2 Report

The authors have made a substantial improvement for this article. The manuscript can be accepted for publishment in the present form.
